# Fairness via In-Processing in the Over-parameterized Regime: A Cautionary Tale with MinDiff Loss

**Akshaj Kumar Veldanda**                                             *akv275@nyu.edu*
*Electrical and Computer Engineering Department*
*New York University*

**Ivan Brugere**[*]                                             *ivan.brugere@jpmchase.com*
*JP Morgan Chase AI Research*

**Jiahao Chen**[*]                                             *cjiahao@gmail.com*
*Parity*

**Sanghamitra Dutta**[*]                                             *sanghamd@umd.edu*
*Electrical and Computer Engineering Department*
*University of Maryland College Park*

**Alan Mishler**[*]                                             *alan.mishler@jpmchase.com*
*JP Morgan Chase AI Research*

**Siddharth Garg**                                             *sg175@nyu.edu*
*Electrical and Computer Engineering Department*
*New York University*

**Reviewed on OpenReview:** *https://openreview.net/forum?id=f4VyYhkRvi&noteId*

## Abstract

Prior work has observed that the test error of state-of-the-art deep neural networks often continues to decrease with increasing over-parameterization, a phenomenon referred to as double descent. This allows deep learning engineers to instantiate large models without having to worry about over-fitting. Despite its benefits, however, prior work has shown that over-parameterization can exacerbate bias against minority subgroups. Several fairness-constrained DNN training methods have been proposed to address this concern. Here, we critically examine MinDiff, a fairness-constrained training procedure implemented within TensorFlow's Responsible AI Toolkit, that aims to achieve Equality of Opportunity. We show that although MinDiff improves fairness for under-parameterized models, it is likely to be ineffective in the over-parameterized regime. This is because an overfit model with zero training loss is trivially group-wise fair on training data, creating an "illusion of fairness," thus turning off the MinDiff optimization (this will apply to any disparity-based measures which care about errors or accuracy; while it won't apply to demographic parity). We find that within specified fairness constraints, under-parameterized MinDiff models can even have lower error compared to their over-parameterized counterparts (despite baseline over-parameterized models having lower error compared to their under-parameterized counterparts). We further show that MinDiff optimization is very sensitive to choice of batch size in the under-parameterized regime. Thus, fair model training using MinDiff requires time-consuming hyper-parameter searches. Finally, we suggest using previously proposed regularization techniques, viz. L2, early stopping and flooding in conjunction with MinDiff to train fair over-parameterized models. In our results, over-parameterized models trained using MinDiff+regularization with standard batch sizes are fairer than their under-parameterized counterparts, suggesting that at the very least, regularizers should be integrated into fair deep learning flows, like MinDiff.

---

[*]Equal Contribution

# 1 Introduction

Over the past few years, machine learning (ML) solutions have found wide applicability in a wide range of domains. However, recent work has shown that ML methods can exhibit unintended biases towards specific population groups, for instance in applications like hiring (Schumann et al., 2020), credit verification (Khandani et al., 2010), facial recognition (Buolamwini & Gebru, 2018; Grother et al., 2010; Ngan & Grother, 2015), recidivism prediction (Chouldechova, 2017) and recommendation systems (Biega et al., 2018; Singh & Joachims, 2018), resulting in negative societal consequences. To address this concern, there is a growing and influential body of work on mitigating algorithmic unfairness of ML models. These solutions are being integrated within widely used ML frameworks and are beginning to find practical deployment (Responsible-AI; Akihiko Fukuchi, 2020). As ML fairness methods make the transition from theory to practice, their ability to achieve stated goals in real-world deployments merits closer examination.

Methods to train fair models can be broadly categorized based on the stage at which they are used: pre-training, in-training, or post-training. Of these, only in-training methods substantively modify the model training process. This paper examines the performance of MinDiff (Prost et al., 2019), the principal in-training method integrated within TensorFlow's Responsible AI Toolkit (Responsible-AI). We are particularly interested in MinDiff for training fair deep learning models because, given TensorFlow's widespread adoption, there is good reason to believe that it will be picked as the default choice by practitioners working within this framework.

We evaluate MinDiff on three datasets, Waterbirds, CelebA, and HAM10000, that are commonly used in fairness literature; and we observe several notes of caution. Because the success of deep learning can be attributed at least in part to the surprising ability of over-parameterized deep networks to generalize (Nakkiran et al., 2020), we begin by evaluating the relationship between model capacity and fairness with MinDiff. Over-parameterized models have more parameters than required to memorize the training dataset; under-parameterized models have fewer. The point at which a model is just large enough to memorize the training data is referred to as the interpolation threshold. We observe that MinDiff does increase fairness for small, under-parameterized models, but is almost entirely ineffective on larger over-parameterized networks. Thus, in some cases, under-parameterized MinDiff models can have lower fairness-constrained error compared to their over-parameterized counterparts even though over-parameterized models are always better on baseline error (i.e., error on models trained without MinDiff optimization). We caution that when using MinDiff for fairness, ML practitioners must carefully choose model capacity, something which is generally unnecessary when fairness is not a concern and the goal is simply to minimize error.

We find the reason MinDiff is ineffective in the over-parameterized regime is because when a model's training loss goes to zero, any fairness metric that relies on differences in the model's accuracy across different sub-groups will also go to zero, thus creating an "illusion of fairness" and turning off MinDiff optimization. For completeness, we note that this argument does not apply to demographic parity (Calders et al., 2009) which cares about equal proportions of positive predictions in each group. Thus, we explore whether strong regularization used along with MinDiff can alleviate its ineffectiveness. Specifically, we consider two classes of regularization techniques: implicit such as batch sizing (Smith et al., 2021; Barrett & Dherin, 2021) and early stopping (Morgan & Bourlard, 1990), and explicit such as weight decay (Krogh & Hertz, 1992) and a recently proposed "loss flooding" method (Ishida et al., 2020) regularizers. We find that: (1) batch sizing only helps for medium sized models around the interpolation threshold; (2) the remaining three methods all improve fairness in the over-parameterized regime; (3) early-stopping and flooding result in the fairest models for the Waterbirds, CelebA and HAM10000 datasets, respectively; and (4) with effective regularization, over-parameterized models are fairer than their under-parameterized counterparts.

# 2 Related Work

There are several techniques in the literature to mitigate algorithmic bias. These techniques can be broadly categorized as: pre-processing, in-processing and post-processing. Pre-processing techniques aim to de-identify sensitive information and create more balanced training datasets (Quadrianto et al., 2019; Ryu et al., 2018; Feldman et al., 2015; Wang & Deng, 2019; Karkkainen & Joo, 2021; Dixon et al., 2018). In-processing techniques (Prost et al., 2019; Cherepanova et al., 2021; Sagawa et al., 2020a;b; Padala & Gujar, 2021;

Agarwal et al., 2018; Zafar et al., 2019; Donini et al., 2018; Lahoti et al., 2020; Beutel et al., 2019; Martinez et al., 2020; Wadsworth et al., 2018; Goel et al., 2018; Wang & Deng, 2019; Hashimoto et al., 2018) alter the training mechanism by imposing fairness constraints to the training objective, or utilize adversarial training (Beutel et al., 2017; Zhang et al., 2018; Madras et al., 2018) to make predictions independent of sensitive attributes. Post-processing techniques (Hardt et al., 2016; Wang et al., 2020; Savani et al., 2020; Chzhen et al., 2019; Jiang et al., 2020; Wei et al., 2020) alter the outputs of an existing model, for instance, using threshold correction (Zhou & Liu, 2006; Collell et al., 2016; Menon et al., 2021a) that applies different classification thresholds to each sensitive group (Hardt et al., 2016). In this paper, we focus on MinDiff (Prost et al., 2019), the primary in-processing procedure implemented within TensorFlow's Responsible AI toolkit. While our quantitative conclusions might differ, we believe that similar qualitative conclusions might hold for other in-processing methods because overfit models are trivially fair.

With the growing adoption of large over-parameterized deep networks, recent efforts have sought to investigate their fairness properties (Menon et al., 2021b; Sagawa et al., 2020a; Cherepanova et al., 2021; Sagawa et al., 2020b). Pham et al. (2021); Dehghani et al. (2023) observed that over-parameterized ERM models have better worst-group generalization compared to their under-parameterized counterparts. However, Maity et al. (2022) warn that baseline ERM models should not be considered state-of-the-art to train fair over-parameterized models. Sagawa et al. (2020b) proposed a pre-processing technique by investigating the role of training data characteristics (such as ratio of majority to minority groups and relative informativeness of spurious versus core features) on fairness and observed that sub-sampling improves fairness in the over-parameterized regime. Menon et al. (2021b) found that post-processing techniques including retraining with sub-sampled majority groups and threshold correction also enhance fairness in over-parameterized models. Alabdulmohsin & Lucic (2021) proposed a post-processing algorithm that can be effectively used to debias large models. Cherepanova et al. (2021) report that in-processing convex surrogates of fairness constraints like equal loss, equalized odds penalty, disparate impact penalty, etc., (Padala & Gujar, 2021) are ineffective on over-parameterized models, but do not propose any techniques to increase the effectiveness of in-processing methods. Wald et al. (2022) theoretically show that interpolating models cannot satisfy fairness constraints. However, it is not thoroughly investigated how fairness constraints can be effectively implemented in over-parameterized models. It is possible that using methods such as MinDiff may improve the training of fair over-parameterized models. Our work is the first to systematically compare under- vs. over-parameterized deep learning models trained using in-processing fairness methods, using MinDiff as a representative technique.

Regularization techniques (Morgan & Bourlard, 1990; Srivastava et al., 2014; Krogh & Hertz, 1992; Ishida et al., 2020) are popularly used in deep learning frameworks to avoid over-fitting. Lately, researchers have also exploited the benefits of regularizers to train fair models. For example, Sagawa et al. (2020a) proposed distributionally robust optimization (DRO) to improve worst-group generalization, but they observed that their approach fails if training loss converges to zero. Hence, they use L2 weight regularization and early stopping to improve fairness in the over-parameterized regime. Our paper systematically evaluates different regularizers, including batch sizing, early stopping, weight decay and the recently proposed flooding loss (Ishida et al., 2020) for MinDiff training across different model sizes, and makes several new observations about the role of regularization in enhancing fairness.

## 3 Methodology

We now describe our evaluation methodology.

### 3.1 Formal Setup

In this paper, we consider binary classification problem on a training dataset $D = \{x_i, a_i, y_i\}_{i=1}^N$, where $x_i$ is an input (an image for instance) $a_i \in \{0, 1\}$ is a sensitive attribute of the input, and $y_i \in \{0, 1\}$ is the corresponding ground-truth label. The training data is sampled from a joint distribution $\mathcal{D}_{X,A,Y}$ over random variables $X$, $A$, and $Y$. Deep neural network (DNN) classifiers are represented as a parameterized function $f_\theta : \mathcal{X} \to [0, 1]$, where $\theta$ are trainable parameters, obtained in practice by minimizing the empirical binary

cross-entropy primary loss function $\mathcal{L}_P$:

$$\mathcal{L}_P = -\frac{1}{N} \sum_{i=1}^{N} [y_i \cdot \log(f_\theta(x_i)) + (1 - y_i) \cdot \log(1 - f_\theta(x_i))] \tag{1}$$

via stochastic gradient descent (SGD).

We denote the classification threshold as $\tau$ which can be used to make predictions $\hat{f}_\theta(x; \tau)$ as shown below

$$\hat{f}_\theta(x; \tau) = \begin{cases} 1, & f_\theta(x) \geq \tau \\ 0, & \text{otherwise.} \end{cases} \tag{2}$$

The goal is to attain low error at the population level, i.e., $\mathbb{P}[\hat{f}_\theta(X; \tau) \neq Y]$ (typically, $\tau = 0.5$). As the data-generating distribution is unavailable, standard DNN training seeks to achieve low empirical error on a test set, which we refer to as *test error* for short. However, performance conditioned on sensitive attributes can vary, leading to outcomes that are biased in favor of or against specific sub-groups. Several fairness metrics have been defined in prior work to account for this bias; in this paper, we will use the widely adopted equality of opportunity metric (Hardt et al., 2016).

**Equality of Opportunity** (Hardt et al., 2016) is a widely adopted fairness notion that seeks to equalize false negative rates (FNR) across sensitive groups to achieve fairness in contexts where a higher FNR for a certain group can result in unfair outcomes, such as wrongful convictions or denial of opportunities in comparison to other groups. For binary sensitive attributes the $\text{FNR}_{\text{gap}}$ at the population level is defined as:

$$\left| \mathbb{P}[\hat{f}_\theta(X; \tau) = 0 | Y = 1, A = 0] - \mathbb{P}[\hat{f}_\theta(X; \tau) = 0 | Y = 1, A = 1] \right|. \tag{3}$$

In practice, we measure the population-level $\text{FNR}_{\text{gap}}$ on a test set, which we will refer to as just $\text{FNR}_{\text{gap}}$ for short. As we describe next, MinDiff (and several other methods) seek to minimize the $\text{FNR}_{\text{gap}}$ during training.

## 3.2 MinDiff Training

MinDiff (Prost et al., 2019) is an in-processing optimization framework that seeks to achieve a balance between two objectives: low test error and low $\text{FNR}_{\text{gap}}$. For this, MinDiff proposes a modified loss function $\mathcal{L}_T = \mathcal{L}_P + \lambda \mathcal{L}_M$, where $\mathcal{L}_T$ is the total loss, that is a weighted sum of the primary cross-entropy loss, defined in Equation (1), and $\mathcal{L}_M$, a differentiable proxy for the $\text{FNR}_{\text{gap}}$. In the modified loss function, $\lambda \in \mathbb{R}_+$ is a user-defined parameter that controls the relative importance of the fairness versus the empirical cross-entropy loss. The fairness term in the modified loss function, $\mathcal{L}_M$, uses the maximum mean discrepancy (MMD) distance between the neural network's outputs for the two sensitive groups when $Y = 1$, i.e.,

$$\mathcal{L}_M = \text{MMD}(f_\theta(X) | A = 0, Y = 1, f_\theta(X) | A = 1, Y = 1). \tag{4}$$

Intuitively, the MMD loss $\mathcal{L}_M$ penalizes any statistical dependence between the predictions for the two subgroups which are made by the model on positive examples. We refer the reader to Appendix A for a formal mathematical definition of the MMD distance (Prost et al., 2019).

## 3.3 Post-hoc Threshold Correction

Due to fairness requirements of the application, ML practitioners might seek to train models with a population level $\text{FNR}_{\text{gap}}$ lower than a specified threshold $\Delta_{\text{FNR}}$. This can be done using an additional post-processing step as proposed by Hardt et al. (2016). The idea is to use different classification thresholds for each sub-group, $\tau_{A=0}$ and $\tau_{A=1}$, that are selected such that the population level $\text{FNR}_{\text{gap}}$ under these thresholds is below the constraint:

$$\left| \mathbb{P}[\hat{f}_\theta(X; \tau_{A=0}) = 0 | Y = 1, A = 0] - \mathbb{P}[\hat{f}_\theta(X; \tau_{A=1}) = 0 | Y = 1, A = 1] \right| \leq \Delta_{\text{FNR}} \tag{5}$$

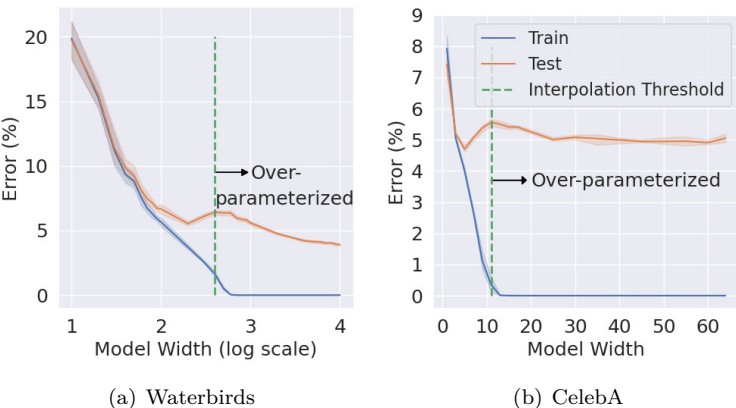

(a) Waterbirds  (b) CelebA

Figure 1: We show the model-wise double descent behaviour on baseline models trained using (a) Waterbirds and (b) CelebA datasets respectively. Interpolation threshold (shown in green dotted line) is the point where the model is large enough to fit the training data. The region beyond the interpolation threshold is called the over-parameterized regime.

In practice, the two thresholds can be picked using grid search and empirically measuring population level $\text{FNR}_{\text{gap}}$ at each grid point on a validation dataset. The resulting thresholds yield a test error which we refer to as the **fairness-constrained test error**. Pareto front of fairness-constrained test error and $\Delta_{\text{FNR}}$ is used to compare different model sizes and fairness methods. In particular, a model or method is fairer if it has lower fairness-constrained test error compared to the alternative for a fixed $\Delta_{\text{FNR}}$.

### 3.4 Regularization Techniques

We evaluate four regularization techniques to improve performance of MinDiff on over-parameterized networks.

**Reduced Batch Sizes:** Due to the stochastic nature of SGD, smaller batch sizes can act as implicit regularizers during training (Smith et al., 2021; Barrett & Dherin, 2021). This is because smaller batch sizes provide a noisy estimate of the total loss $\mathcal{L}_T$.

**Weight Decay:** Weight decay (Krogh & Hertz, 1992) explicitly penalizes the parameters $\theta$ of the DNN from growing too large. Weight decay adds a penalty, usually the L2 norm of the weights, to the loss function.

**Early Stopping:** Early stopping (Morgan & Bourlard, 1990) terminates DNN training earlier than the point at which training loss converges to a local minima, and has been shown to be particularly effective for over-parameterized deep networks (Li et al., 2019). A common implementation of early stopping is to terminate training once the validation loss has not improved for a certain number of gradient steps or epochs (Morgan & Bourlard, 1990). For models trained with MinDiff, we explore two versions of early stopping in which we use either the primary loss, $\mathcal{L}_P$, or total loss, $\mathcal{L}_T$, as a stopping criterion.

**Flooding Regularizer:** Finally, motivated by our goal to prevent the primary training loss from going to zero (which then turns off MinDiff as well), we apply the flooding regularizer (Ishida et al., 2020) that encodes this goal *explicitly*. Flooding operates by performing gradient descent only if $\mathcal{L}_P > b$, where $b$ is the flood level. Otherwise, if $\mathcal{L}_P \leq b$, then gradient ascent takes place as shown in Equation 6. This phenomenon ensures that $\mathcal{L}_P$ floats around the flood level $b$ and never approaches zero. We implement flooding by replacing the primary loss term in $\mathcal{L}_T$ with a new loss term:

$$\mathcal{L}'_P = \left| \mathcal{L}_P - b \right| + b, \tag{6}$$

which, in turn, enables continued minimization of the MinDiff loss term, $\mathcal{L}_M$, over the training process.

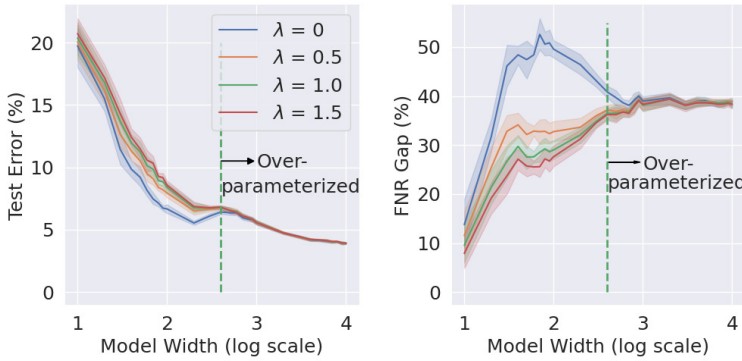

Figure 2: We show the (a) average test error, and (b) $\text{FNR}_{\text{gap}}$ versus model width for MinDiff optimization with $\lambda = \{0.0, 0.5, 1.0, 1.5\}$'s on Waterbirds dataset. MinDiff optimization has negligible to no impact on fairness in the over-parameterized models. However, for under-parameterized models, we find that increasing $\lambda$ substantially reduces the $\text{FNR}_{\text{gap}}$.

## 4 Experimental Setup

We perform our experiments on the Waterbirds (Sagawa et al., 2020a), CelebA (Liu et al., 2015) and HAM10000 (Tschandl, 2018) datasets which have previously been used in fairness evaluations of deep learning models. Here, we describe network architectures, training and evaluations for the two datasets.

### 4.1 Waterbirds Dataset

Waterbirds is a synthetically created dataset (Sagawa et al., 2020a) which contains water- and land-bird images overlaid on water and land backgrounds. A majority of waterbirds (landbirds) appear in water (land) backgrounds, but in a minority of cases waterbirds (landbirds) also appear on land (water) backgrounds. As in past work, we use the background as the sensitive feature. Further, we use waterbirds as the positive class and landbirds as the negative class. The dataset is split into training, validation and test sets with 4795, 1199 and 5794 images in each dataset respectively.

We follow the training methodology described in (Sagawa et al., 2020b) to train a deep network for this dataset. First, a fixed pre-trained ResNet-18 model is used to extract a $d$-dimensional feature vector $\mu$. This feature vector is then converted into an $m$-dimensional feature $\mu' = ReLU(U\mu)$, where $U \in \mathbb{R}^{m \times d}$ is a random matrix with each row sampled uniformly from a unit sphere $\mathcal{S}^{d-1}$. A logistic regression classifier is trained on $\mu'$. Model width is controlled by varying $m$, the dimensionality of $\mu'$, from 10 to 10,000.

### 4.2 CelebA Dataset

The CelebA dataset consists of 202,599 celebrity face images annotated with 40 binary attributes including gender, hair colour, hair style, eyeglasses, etc. In our experiments, we set the target label $\mathcal{Y}$ to be hair color, which is either blond ($Y = 1$) or non-blond ($Y = 0$), and the sensitive attribute to be gender. Blond individuals constitute only 15% of this dataset, and only 6% of blond individuals are men. Consequently baseline models make disproportionately large errors on blond men versus blond women. The objective of MinDiff training is to minimize the $\text{FNR}_{\text{gap}}$ between blond men and blond women[1]. The dataset is split into training, validation and test sets with 162770, 19867 and 19962 images, respectively.

We used the ResNet-preact-18 model for this dataset, and vary model capacity by uniformly scaling the number of channels in all layers. We also train a LeNet-5 architecture on CelebA dataset and compare it with the ResNet-preact-18 model.

---

[1]Note that the MinDiff paper uses False Positive Rate, but this is totally arbitrary here since the class labels are arbitrary.

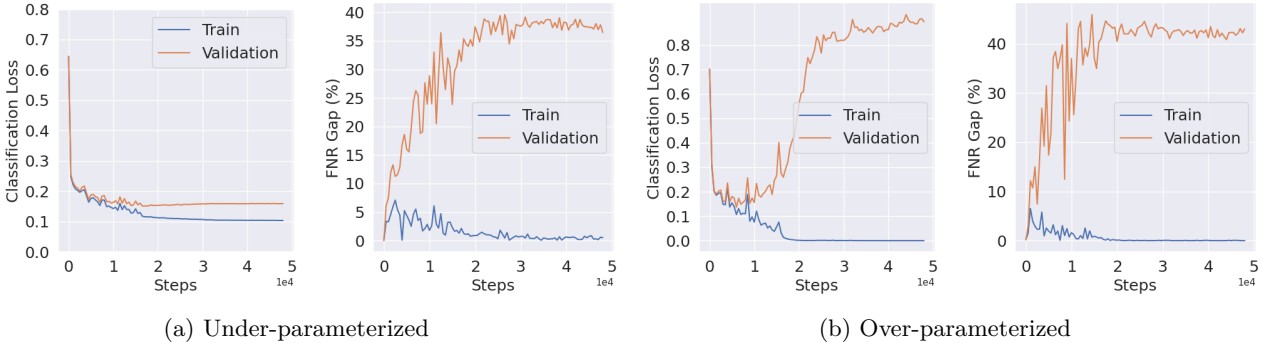

(a) Under-parameterized

(b) Over-parameterized

Figure 3: We show the progress of primary loss and population level $FNR_{gap}$, evaluated on training dataset and on the validation dataset, versus SGD steps during MinDiff optimization ($\lambda = 1.5$) for under-parameterized and over-parameterized models on CelebA dataset. We find that over-parameterized models over-fits to the training data and achieve zero population level $FNR_{gap}$ on the training dataset, thus turning off MinDiff optimization. Whereas, population level $FNR_{gap}$ on the training dataset is positive in the under-parameterized models, allowing for MinDiff optimization to be effective.

### 4.3 HAM10000 Dataset

HAM10000 dataset consists of 10,015 dermatoscopic images for automatic diagnosis of pigmented skin lesions. Each image in the dataset contains several annotations including sensitive attributes like age, gender and the diagnosis labels of the patient. Medical imaging is a problem with direct human impact and this dataset has been previously studied in the context of bias and fairness (Daneshjou et al., 2022). Following prior work (Zong et al., 2023; Maron et al., 2019), we split the 7 diagnostic labels into benign: basal cell carcinoma (bcc), benign keratosis-like lesions (bkl), dermatofibroma (df), melanocytic nevi (nv), and vascular lesions (vasc), and malignant: actinic keratoses and intraepithelial carcinoma / Bowen's disease (akiec), and melanoma (mel). In our experiments, we set the target label $\mathcal{Y}$ to be the diagnostic label, which is either malignant ($Y = 1$) or benign ($Y = 0$), and the sensitive attribute to be age. We categorize all patients into two demographic groups based on their age: those under 40 ($< 40$) and those 40 years old or above ($\geq 40$). Malignant individuals constitute only 15% of this dataset, and only 6% of malignant individuals are $< 40$. Consequently baseline models make disproportionately large errors on malignant ($< 40$) individuals versus malignant ($\geq 40$) individuals. The objective of MinDiff training is to minimize the $FNR_{gap}$ between malignant ($< 40$) and malignant ($\geq 40$) individuals. We discard images whose sensitive attributes are not available and the dataset is split into training, validation and test sets with 7967, 989 and 1002 images, respectively. We used the pre-trained ResNet-18 as an over-parameterized model and LeNet-5 as an under-parameterized model for this dataset.

### 4.4 Hyper-parameters and Training Details

**Waterbirds**   We train for a total of $30,000$ gradient steps using the Adam optimizer. For our baseline experiments, we set batch size to 128 and use a learning rate schedule with initial learning rate = 0.01 and decay factor of 10 for every $10,000$ gradient steps. We ran every experiment 10 times with random initializations and report the average of all the runs. We trained and evaluated all the models using the Waterbirds dataset on an Intel Xeon Platinum 8268 CPU (24 cores, 2.9 GHz).

**CelebA**   We train for a total of $48,000$ gradient steps using the Adam optimizer. We set the baseline batch size to 128 and adopt a learning rate scheduler with initial learning rate of 0.0001 and decay factor of 10 for every $16,000$ gradient steps. In our experiments, we varied the number of channels in the first ResNet-preact-18 block from 1 to 64 (the number of channels is then scaled up by two in each block). We trained all the models using the CelebA dataset on NVIDIA 4 x V100 (32 GB) GPU cards.

**HAM10000**  We train for a total of $12,600$ and $7,000$ gradient steps using the Adam optimizer for LeNet-5 and ResNet-18 architectures, respectively. We set the baseline batch size to 128 and set the learning rate to 0.0001. We adopt a learning rate scheduler with a decay factor of 10 for every $1,260$ gradient steps for ResNet-18 models. We trained all the models using the HAM10000 dataset on NVIDIA V100 (32 GB) GPU cards.

For Waterbirds and CelebA datasets, we performed MinDiff training with values of $\lambda = \{0.0, 0.5, 1.0, 1.5\}$, where recall that $\lambda$ controls the importance of the fairness objective. $\lambda = 0.0$ corresponds to training with the primary loss only, and we refer to the resulting model as the baseline model. To study the effect of batch sizing, we trained additional models with batch sizes $\{8, 32\}$. We explored with three different weight decay strengths $= \{0.001, 0.1, 10.0\}$ and two different flood levels, $b \in \{0.05, 0.1\}$. We report the average $\pm\ 95\%$ confidence interval in all figures and tables. For HAM10000 dataset, we performed MinDiff training with values of $\lambda = \{0.0, 0.5, 1.0, 1.5\}$. We explored with two different weight decay strengths $= \{0.001, 0.1\}$ and $= \{0.001, 0.01\}$ for LeNet-5 and ResNet-18 architectures, respectively. Additionally, we tested the impact of two different flood levels, which were $b \in \{0.2, 0.3\}$ for LeNet-5 and $b \in \{0.05, 0.1\}$ for ResNet-18.

## 5 Experimental Results

### 5.1 Identifying the Interpolation Threshold

To distinguish between under- and over-parameterized models, we begin by identifying the interpolation threshold; the point where the model is sufficiently large to interpolate the training data (achieve population level zero error on training data). Figure 1(a) and Figure 1(b) show the population level error on training data and test error curves for baseline training versus model width for the Waterbirds and CelebA datasets, respectively, showing interpolation thresholds at model widths of 400 for Waterbirds and 11 for CelebA. On both the datasets, we also observe the double descent phenomenon (Nakkiran et al., 2020), where the test error *decreases* with increasing model capacity beyond the interpolation threshold. That is, for the original model (training without MinDiff), the largest models provide high accuracy.

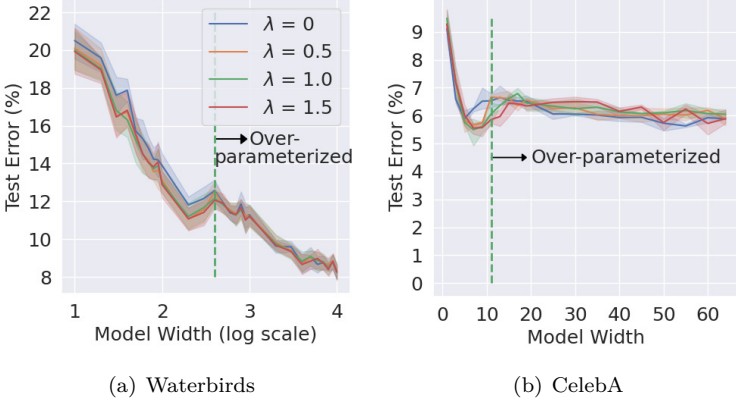

(a) Waterbirds

(b) CelebA

Figure 4: We plot the fairness constrained test error with a $\Delta_{\mathrm{FNR}} \leq 10\%$ constraint on the MinDiff trained models with $\lambda = \{0.0, 0.5, 1.0, 1.5\}$ on (a) Waterbirds and (b) CelebA datasets respectively. On both these datasets, we find that MinDiff is only effective for under-parameterized models.

### 5.2 MinDiff Evaluation

We now re-train our models with MinDiff optimization with $\lambda = \{0.0, 0.5, 1.0, 1.5\}$. Figure 2 shows the test error and $\mathrm{FNR}_{\mathrm{gap}}$ versus model width for the Waterbirds dataset. In the under-parameterized regime, we observe that, increasing the MinDiff weights significantly reduces the $\mathrm{FNR}_{\mathrm{gap}}$ with only a small drop in test accuracy. However, in the over-parameterized regime, we find that MinDiff training has no impact on either test error or the $\mathrm{FNR}_{\mathrm{gap}}$.

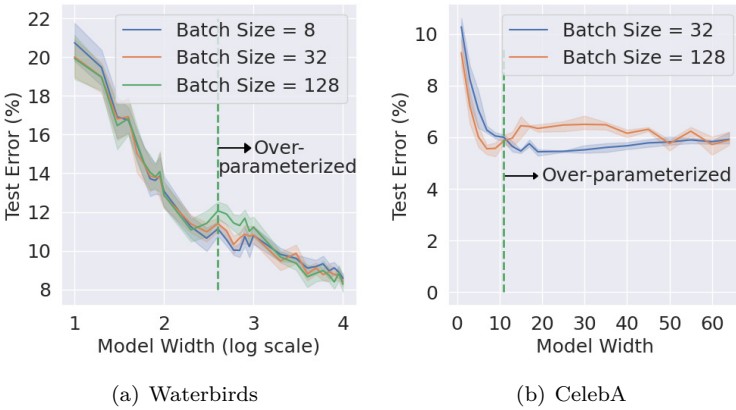

(a) Waterbirds      (b) CelebA

Figure 5: We plot the fairness constrained test error with a $\Delta_{\text{FNR}} \leq 10\%$ constraint on the MinDiff trained model with $\lambda = 1.5$ on (a) Waterbirds and (b) CelebA datasets respectively. We find that smaller batch sizes are only effective around the interpolation threshold.

Table 1: We pick three under-parameterized and four over-parameterized model widths and report the average test error and $\text{FNR}_{\text{gap}}$ for baseline training and several values of $\lambda$ on CelebA dataset. We find that, in the under-parameterized models, the drop in $\text{FNR}_{\text{gap}}$ for MinDiff training vs baseline training is more compared to that in the over-parameterized models. We report each data point by averaging over 10 runs.

| Width | $\lambda = 0$ | | $\lambda = 0.5$ | | $\lambda = 1.0$ | | $\lambda = 1.5$ | |
|---|---|---|---|---|---|---|---|---|
| | Error | FNR Gap | Error | FNR Gap | Error | FNR Gap | Error | FNR Gap |
| 5 (Under-) | $4.74 \pm 0.07$ | $48.77 \pm 1.44$ | $5.55 \pm 0.39$ | $39.85 \pm 3.82$ | $5.46 \pm 0.36$ | $40.45 \pm 5.58$ | $5.47 \pm 0.34$ | $37.06 \pm 5.88$ |
| 7 (Under-) | $5.07 \pm 0.07$ | $47.35 \pm 1.53$ | $5.76 \pm 0.5$ | $40.27 \pm 2.45$ | $5.64 \pm 0.52$ | $42.7 \pm 3.12$ | $5.48 \pm 0.4$ | $43.05 \pm 4.2$ |
| 9 (Under-) | $5.31 \pm 0.07$ | $44.78 \pm 1.72$ | $5.72 \pm 0.4$ | $42.37 \pm 1.75$ | $5.77 \pm 0.53$ | $40.91 \pm 3.28$ | $5.52 \pm 0.42$ | $42.43 \pm 3.74$ |
| 13 (Over-) | $5.52 \pm 0.07$ | $42.79 \pm 2.03$ | $5.57 \pm 0.1$ | $42.16 \pm 2.33$ | $5.66 \pm 0.17$ | $42.53 \pm 0.98$ | $5.67 \pm 0.32$ | $43.07 \pm 2.12$ |
| 19 (Over-) | $5.36 \pm 0.09$ | $43.98 \pm 1.81$ | $5.33 \pm 0.05$ | $42.52 \pm 1.25$ | $5.42 \pm 0.07$ | $42.63 \pm 1.01$ | $5.45 \pm 0.11$ | $41.34 \pm 1.41$ |
| 55 (Over-) | $4.98 \pm 0.07$ | $44.14 \pm 1.55$ | $5.05 \pm 0.07$ | $42.46 \pm 1.28$ | $5.24 \pm 0.06$ | $42.65 \pm 1.85$ | $5.47 \pm 0.15$ | $41.93 \pm 1.88$ |
| 64 (Over-) | $4.99 \pm 0.06$ | $42.54 \pm 2.01$ | $5.11 \pm 0.06$ | $43.12 \pm 2.04$ | $5.28 \pm 0.12$ | $41.8 \pm 2.32$ | $5.40 \pm 0.15$ | $41.56 \pm 2.7$ |

Table 1 shows the test error and $\text{FNR}_{\text{gap}}$ for selected under- and over-parameterized models on the CelebA dataset trained with MinDiff. Our conclusions are qualitatively the same as for Waterbirds. We find that, MinDiff training significantly reduces the $\text{FNR}_{\text{gap}}$ compared to the baseline model in the under-parameterized regime, for example, from a 48% FNR gap without MinDiff to a 37% FNR gap with $\lambda = 1.5$ for a model width of 5. On the other hand, MinDiff training compared with the baseline model has a negligible effect on both test error and $\text{FNR}_{\text{gap}}$ in the over-parameterized regime, sometimes even resulting in a (small) increase in both.

Table 2: We tabulate the fairness constrained test error (with a $\Delta_{\text{FNR}} \leq 10\%$ constraint) of early stopped MinDiff models (trained with $\lambda = \{0.5, 1.0, 1.5\}$) for different widths. We compare two methods for early stopping (es) based on the stopping criterion: primary validation loss ($\text{es}(\mathcal{L}_P)$) and total validation loss ($\text{es}(\mathcal{L}_T)$). We find that, for CelebA dataset, MinDiff + $\text{es}(\mathcal{L}_P)$ is better than MinDiff + $\text{es}(\mathcal{L}_T)$.

| | $\lambda = 0.5$ | | | $\lambda = 1.0$ | | | $\lambda = 1.5$ | | |
|---|---|---|---|---|---|---|---|---|---|
| | Width = 5 | Width = 19 | Width = 64 | Width = 5 | Width = 19 | Width = 64 | Width = 5 | Width = 19 | Width = 64 |
| No es | $6.36 \pm 0.51$ | $6.54 \pm 0.12$ | $5.88 \pm 0.13$ | $6.25 \pm 0.43$ | $6.43 \pm 0.12$ | $6.06 \pm 0.06$ | $6.36 \pm 0.35$ | $6.54 \pm 0.18$ | $6.07 \pm 0.2$ |
| $\text{es}(\mathcal{L}_P)$ | $6.17 \pm 0.34$ | $5.91 \pm 0.26$ | $5.45 \pm 0.22$ | $6.32 \pm 0.28$ | $5.82 \pm 0.32$ | $4.79 \pm 0.17$ | $6.37 \pm 0.25$ | $5.63 \pm 0.3$ | $4.70 \pm 0.07$ |
| $\text{es}(\mathcal{L}_T)$ | $7.76 \pm 0.56$ | $6.47 \pm 0.31$ | $5.76 \pm 0.61$ | $7.79 \pm 0.47$ | $7.18 \pm 0.58$ | $6.05 \pm 0.53$ | $8.65 \pm 0.65$ | $7.38 \pm 0.77$ | $6.80 \pm 0.64$ |

We observe that MinDiff performs poorly for over-parameterized models: Figure 3 shows how the population level primary loss and $\text{FNR}_{\text{gap}}$'s change, evaluated on training and validation datasets, during SGD steps for under-parameterized and over-parameterized models on the CelebA dataset. We note that at the beginning of training, the population level $\text{FNR}_{\text{gap}}$ on training dataset is small because randomly initialized models make random predictions. These random predictions are fair w.r.t the error rates but not necessarily fair according

to other criteria like Demographic Parity. As training progresses, the over-parameterized model eventually over-fits the training data at around 20,000 steps, achieving zero primary loss. This model also appears to be trivially fair from the standpoint of the training data—observe that at the same point, the population level $\text{FNR}_{\text{gap}}$ on the training dataset also goes to zero, and no further optimization takes place. On the other hand, the population level $\text{FNR}_{\text{gap}}$ during training remains positive for the under-parameterized model.

Figure 4 plots the fairness constrained test error with a $\Delta_{\text{FNR}} \leq 10\%$ constraint using post-training threshold correction on the MinDiff trained models. We can again observe that MinDiff is only effective for under-parameterized models. We include results for larger $\lambda \in \{8.0, 16.0, 32.0\}$ values in Appendix B and observe that the findings do not change. For CelebA, the lowest fairness constrained test error is actually achieved by an under-parameterized model. In other words, *achieving fairness via MinDiff optimization requires careful selection of model width, including exploring the under-parameterized regime.*

Table 3: We tabulate the fairness constrained test error (with a $\Delta_{\text{FNR}} \leq 10\%$ constraint) for different regularization schemes used in conjunction with MinDiff optimization ($\lambda = 1.5$) on LeNet-5 and ResNet-preact-18 (with three model widths $\in \{5, 19, 64\}$) models trained using CelebA dataset. The performance of best regularizer is highlighted in bold for each model width. Notation: wd is weight decay, $\text{es}(\mathcal{L}_P)$ is early stopping w.r.t primary loss and fl is flooding.

| Method | LeNet-5 | Width = 5 | Width = 19 | Width = 64 |
|---|---|---|---|---|
| | Under- | Under- | Over- | Over- |
| $\lambda = 0$ | $9.73 \pm 0.26$ | $5.92 \pm 0.19$ | $6.51 \pm 0.11$ | $5.76 \pm 0.1$ |
| $\lambda = 1.5$ | $9.53 \pm 0.17$ | $6.36 \pm 0.35$ | $6.54 \pm 0.18$ | $6.07 \pm 0.2$ |
| $\lambda = 1.5 + \text{wd} = 0.001$ | $9.87 \pm 0.28$ | $\mathbf{5.82 \pm 0.2}$ | $6.53 \pm 0.16$ | $5.08 \pm 0.15$ |
| $\lambda = 1.5 + \text{wd} = 0.1$ | $12.68 \pm 0.03$ | $6.15 \pm 0.15$ | $5.82 \pm 0.16$ | $6.57 \pm 0.1$ |
| $\lambda = 1.5 + \text{es}(\mathcal{L}_P)$ | $\mathbf{9.44 \pm 0.14}$ | $6.37 \pm 0.25$ | $5.63 \pm 0.3$ | $4.70 \pm 0.07$ |
| $\lambda = 1.5 + \text{fl} = 0.05$ | $9.53 \pm 0.17$ | $6.28 \pm 0.25$ | $6.30 \pm 0.23$ | $5.49 \pm 0.17$ |
| $\lambda = 1.5 + \text{fl} = 0.1$ | $9.52 \pm 0.17$ | $6.30 \pm 0.3$ | $\mathbf{5.25 \pm 0.25}$ | $\mathbf{4.67 \pm 0.12}$ |

## 5.3 Impact of Regularization in Over-parameterized Regime

We now examine if additional regularization can help improve the fairness of MinDiff-regularized models in the over-parameterized regime. Unless otherwise stated, in all subsequent evaluations we perform post-training threshold correction with a $\Delta_{\text{FNR}} \leq 10\%$ constraint and compare fairness constrained test error.

**Batch sizing only helps around the interpolation threshold.** Small batch sizes cause primary training loss curves to converge more slowly, potentially providing more opportunity for MinDiff optimizations. In Figure 5(a) and Figure 5(b), we plot fairness constrained test error curves versus model widths for different batch sizes on the Waterbirds and CelebA datasets, respectively. We find that smaller batch sizes improve fairness constrained test error only around the interpolation threshold for both the datasets, but do not noticeably benefit smaller and larger models. On further examination, we note the benefits around the interpolation threshold are because smaller batch sizes induce stronger regularization effects and push the interpolation threshold to the right (see Appendix Figure 13). As a result, MinDiff is effective on a slightly increased range of model widths. However, other than this behaviour, we see no other benefits of using batch sizing as a regularizer and do not explore it further.

**Early stopping criterion.** We evaluate two methods for early stopping. The first uses the primary validation loss (MinDiff+$\text{es}(\mathcal{L}_P)$) as a stopping criterion, while the second uses total validation loss for stopping (MinDiff+$\text{es}(\mathcal{L}_T)$). Figure 6 plots fairness constrained test error versus model width for these two schemes and different values of $\lambda$ on Waterbirds, and Table 2 shows the same data for CelebA. We find that both schemes improve fairness for over-parameterized models, but have limited impact in the under-parameterized regime. For Waterbirds, the differences between the two are small, although using primary validation loss as the stopping criterion (MinDiff+$\text{es}(\mathcal{L}_P)$) is marginally better than using total validation loss (MinDiff+$\text{es}(\mathcal{L}_T)$). However, for CelebA, we find that primary loss stopping criterion (MinDiff+$\text{es}(\mathcal{L}_P)$) is substantially better than total loss stopping criterion (MinDiff+$\text{es}(\mathcal{L}_T)$), especially for large models. Thus, we use the former for the remainder of our experiments.

**Comparing regularization methods on Waterbirds.** In Figure 7, we plot the fairness constrained test error versus model width for different regularization schemes including early stopping ($\lambda$+es), weight decay with two different values ($\lambda$+wd=0.001, $\lambda$+wd=0.1) and flooding ($\lambda$+fl) on Waterbirds. We find that the early stopping and weight decay regularizes substantially improve fairness for models just below the interpolation threshold and all over-parameterized models. In contrast, flooding shows only small improvements in fairness. For $\lambda = 0.5$, we find that early stopping is the best across the board. For $\lambda = 1.5$, either early stopping and weight decay are the best depending on model width, although the differences are small.

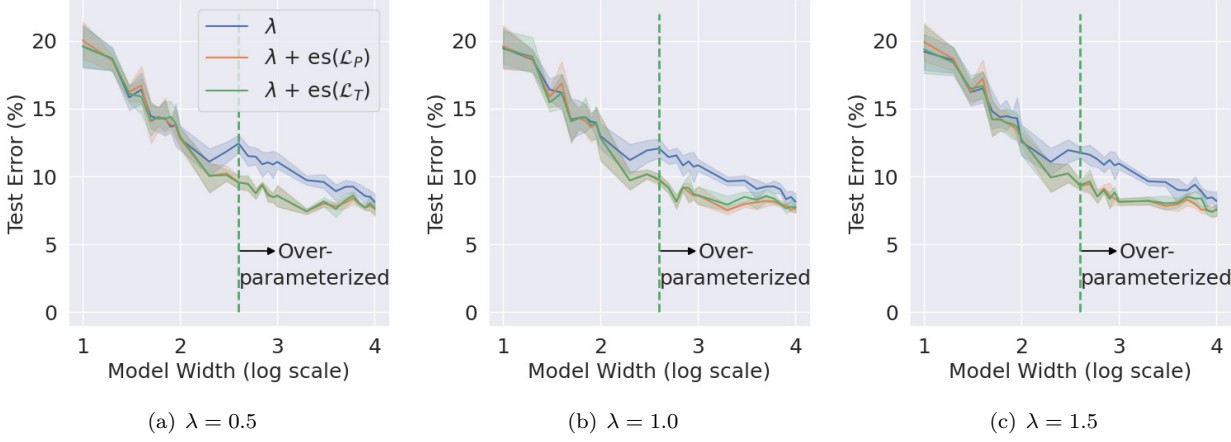

Figure 6: We plot the fairness constrained test error (with a $\Delta_{\text{FNR}} \leq 10\%$ constraint) of early stopped MinDiff models (trained with $\lambda = \{0.5, 1.0, 1.5\}$) for several model widths. We compare two methods for early stopping (es) based on the stopping criterion: primary validation loss (es($\mathcal{L}_P$)) and total validation loss (es($\mathcal{L}_T$)). We find that, for Waterbirds dataset, using either stopping criterion will significantly improve fairness constrained error, especially in the over-parameterized regime.

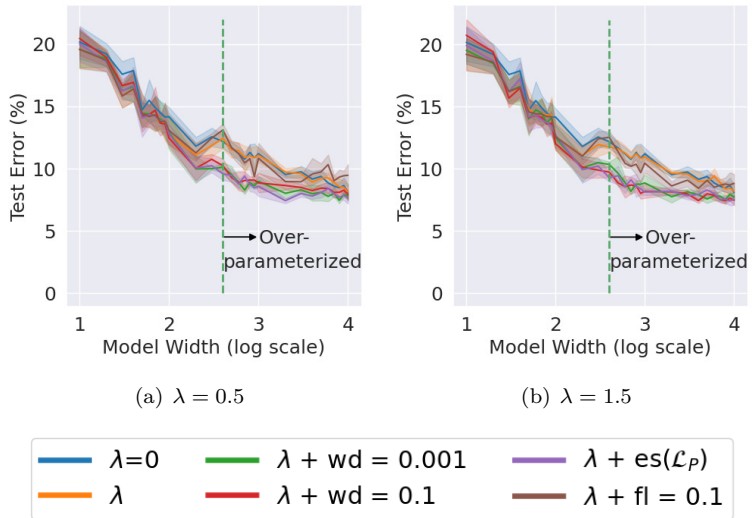

Figure 7: We plot fairness constrained test error (with a $\Delta_{\text{FNR}} \leq 10\%$ constraint) versus model widths for different regularization schemes used in conjunction with MinDiff optimization on Waterbirds dataset. We find that early stopping and weight decay are preferred choice of regularizers for Waterbirds dataset. Notation: wd is weight decay, es($\mathcal{L}_P$) is early stopping w.r.t primary loss and fl is flooding.

Table 4: We tabulate the fairness constrained test error (with a $\Delta_{\text{FNR}} \leq 10\%$ constraint) for different regularization schemes used in conjunction with MinDiff optimization ($\lambda = 0.5$, $\lambda = 1.0$ and $\lambda = 1.5$) on HAM10000 dataset. The performance of best regularizer is highlighted in bold for each model. Notation: wd is weight decay, es($\mathcal{L}_P$) is early stopping w.r.t primary loss, es($\mathcal{L}_T$) is early stopping w.r.t total loss and fl is flooding.

| | $\lambda = 0.5$ | | $\lambda = 1.0$ | | $\lambda = 1.5$ | |
| | LeNet-5 | ResNet-18 | LeNet-5 | ResNet-18 | LeNet-5 | ResNet-18 |
|---|---|---|---|---|---|---|
| $\lambda = 0$ | $14.64 \pm 0.40$ | $\mathbf{10.85 \pm 0.22}$ | $14.64 \pm 0.40$ | $10.85 \pm 0.22$ | $14.64 \pm 0.40$ | $10.85 \pm 0.22$ |
| $\lambda$ | $14.60 \pm 0.35$ | $11.91 \pm 0.12$ | $14.34 \pm 0.74$ | $10.86 \pm 0.45$ | $14.61 \pm 0.53$ | $11.47 \pm 0.19$ |
| $\lambda + \text{wd} = 0.001$ | $\mathbf{13.97 \pm 0.14}$ | $12.77 \pm 1.02$ | $14.60 \pm 0.47$ | $12.34 \pm 0.46$ | $15.01 \pm 0.64$ | $\mathbf{10.75 \pm 0.08}$ |
| $\lambda + \text{wd} = 0.01$ | - | $10.94 \pm 1.29$ | - | $12.74 \pm 1.23$ | - | $12.35 \pm 0.66$ |
| $\lambda + \text{wd} = 0.1$ | $14.27 \pm 0.10$ | - | $\mathbf{14.11 \pm 0.08}$ | - | $14.18 \pm 0.24$ | - |
| $\lambda + \text{es}(\mathcal{L}_P)$ | $14.30 \pm 0.47$ | $13.41 \pm 1.32$ | $15.10 \pm 1.14$ | $12.04 \pm 1.09$ | $14.73 \pm 1.11$ | $13.12 \pm 1.31$ |
| $\lambda + \text{es}(\mathcal{L}_t)$ | $14.20 \pm 0.09$ | $12.54 \pm 0.77$ | $14.14 \pm 0.09$ | $12.51 \pm 0.86$ | $\mathbf{14.15 \pm 0.16}$ | $12.95 \pm 1.4$ |
| $\lambda + \text{fl} = 0.05$ | - | $12.11 \pm 0.37$ | - | $\mathbf{10.68 \pm 0.59}$ | - | $11.02 \pm 0.79$ |
| $\lambda + \text{fl} = 0.1$ | - | $11.78 \pm 1.07$ | - | $12.64 \pm 1.13$ | - | $12.82 \pm 0.51$ |
| $\lambda + \text{fl} = 0.2$ | $14.47 \pm 0.22$ | - | $14.27 \pm 0.65$ | - | $14.53 \pm 0.56$ | - |
| $\lambda + \text{fl} = 0.3$ | $14.77 \pm 0.72$ | - | $14.50 \pm 0.64$ | - | $15.13 \pm 0.89$ | - |

**Comparing regularization methods on CelebA.** Table 3 compares different regularization schemes for $\lambda = 1.5$ on LeNet-5 architecture and three ResNet-preact-18 model widths, each trained on CelebA dataset. For LeNet-5, we observe that early stopping results in lowest fairness constrained test error. For ResNet-preact-18 with smallest width, we find that weight decay schemes result in the lowest fairness constrained test error, while for the large over-parameterized model, flooding works best. Comparing across all models, we find that, the largest model with flooding provides the overall lowest fairness constrained test error. Recalling that flooding was ineffective on Waterbirds, we conclude that no one regularizer works best across datasets and model widths, but additional regularization in general can restore the benefits of over-parameterization with MinDiff training.

**Comparing regularization methods on HAM10000.** From Appendix Figure 9, we observe that for ResNet-18 ($\lambda = 1.5$) the primary loss over-fits to the training dataset and achieves zero training loss thus turning off MinDiff optimization on the over-parameterized model. In contrast, for the under-parameterized model, MinDiff optimization is active and aids in improving fairness since training loss is non-zero. Table 4 compares different regularization schemes on the HAM10000 dataset for $\lambda = 0.5$, $\lambda = 1.0$ and $\lambda = 1.5$. We can observe that the fairness constrained test error of MinDiff model is lower for LeNet-5 architecture compared to baseline model, whereas, for over-parameterized ResNet-18 model, the fairness constrained test error of baseline model is lower compared to MinDiff model. However, for both the models, regularization improves the fairness constrained test error for all values of $\lambda$ except for ResNet-18, where a smaller value of $\lambda = 0.5$ seems to be having no effect on fairness.

## 6    Conclusion

In this paper, we have critically examined the performance of MinDiff, an in-training fairness regularization technique implemented within TensorFlow's Responsible AI toolkit, with respect to DNN model complexity, with a particular eye towards over-parameterized models. We find that although MinDiff improves the fairness of under-parameterized models relative to baseline, it is ineffective in improving fairness for over-parameterized models. As a result, we find that for one of our datasets, under-parameterized MinDiff models have lower fairness constrained test error than their over-parameterized counterparts, suggesting that time-consuming searches for best model size might be necessary when MinDiff is used with the goal of training a fair model.

To address these concerns, we explore traditional batch sizing, weight decay and early stopping regularizers to improve MinDiff training, in addition to flooding, a recently proposed method that is evaluated for the first time in the context of fair training. We find that batch sizing is ineffective in improving fairness except for model widths near the interpolation threshold. The other regularizers do improve fairness for

over-parameterized models, but the best regularizer depends on the dataset and model size. In particular, flooding results in the fairest models on the CelebA dataset, suggesting its utility for MinDiff optimization. Finally, we show that with appropriate choice of regularizer, over-parameterized MinDiff models regain their benefits over under-parameterized counterparts even from a fairness lens.

## Availability

Code with README.txt file is available at: `https://github.com/akshajkumarv/MinDiff`

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

## A    Maximum Mean Discrepancy

Maximum Mean Discrepancy (MMD) is a statistic to test the statistical dependency between two sample distributions. It's mathematical formulation consists of taking the mean between two samples $Z_0$ (consisting of $m$ elements) and $Z_1$ (consisting of $n$ elements) mapped into a Reproducing Kernel Hilbert Subspace, and is computed as follows (Prost et al., 2019):

$$\text{MMD}(Z_0, Z_1) = \frac{1}{m^2} \sum_{i,j=1}^{m} k(z_{0,i}, z_{0,j}) - \frac{2}{mn} \sum_{i,j=1}^{m,n} k(z_{0,i}, z_{1,j}) + \frac{1}{n^2} \sum_{i,j=1}^{n} k(z_{1,i}, z_{1,j}) \tag{7}$$

where $k$ is either a Gaussian or a Laplace kernel, $(z_{0,i})_{i=1,m}$ and $(z_{1,i})_{i=1,n}$ are elements from $Z_0$ and $Z_1$, respectively.

## B    MinDiff Evaluation

To study the sensitivity analysis on $\lambda$, we plot the fairness constrained test error (with a $\Delta_{\text{FNR}} \leq 10\%$ constraint) on Waterbirds dataset for different values of $\lambda = \{0, 1.5, 8.0, 16.0, 32.0\}$ in Figure 8. We observe that MinDiff is only effective for under-parameterized models even for larger values of $\lambda$, but provides no significant gains compared to smaller values of $\lambda$. We also observe that when $\lambda$ is set to high values, like $\lambda = 8.0$ or $16.0$ or $32.0$, the optimization prioritizes minimizing $\text{FNR}_{\text{gap}}$ rather than accurately classifying the data, leading to poor classification performance in the over-parameterized regime. Therefore, it is crucial to choose lambda carefully in order to strike a balance between classification accuracy and fairness for different model widths.

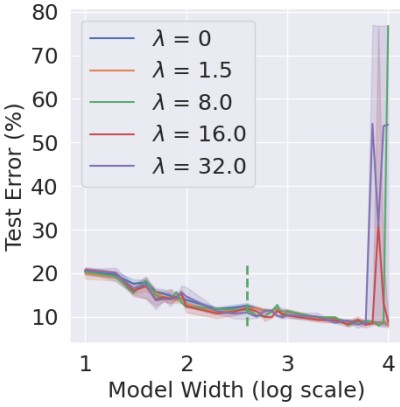

Figure 8: We plot the fairness constrained test error with a $\Delta_{\text{FNR}} \leq 10\%$ constraint on the MinDiff trained models with $\lambda = \{0.0, 1.5, 8.0, 16.0, 32.0\}$ on Waterbirds dataset.

On HAM10000 dataset, we observe that for Resnet-18 the primary loss over-fits to the training dataset and achieves zero training loss thus turning off MinDiff optimization on the over-parameterized model. In contrast, for the under-parameterized model, MinDiff optimization is active and aids in improving fairness since training loss is non-zero.

## C    Impact of Regularization in Over-parameterized Regime

### C.1    Batch Size

### C.1.1    Waterbirds

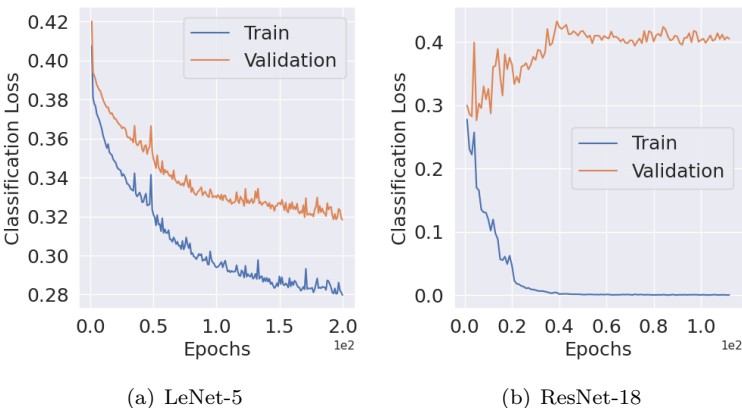

(a) LeNet-5             (b) ResNet-18

Figure 9: We show the progress of primary loss, evaluated on training dataset and on the validation dataset, versus epochs during MinDiff optimization ($\lambda = 1.5$) for under-parameterized (LeNet-5) and over-parameterized (ResNet-18) models on HAM10000 dataset. We find that over-parameterized ResNet-18 model over-fits to the training data, thus turning off MinDiff optimization.

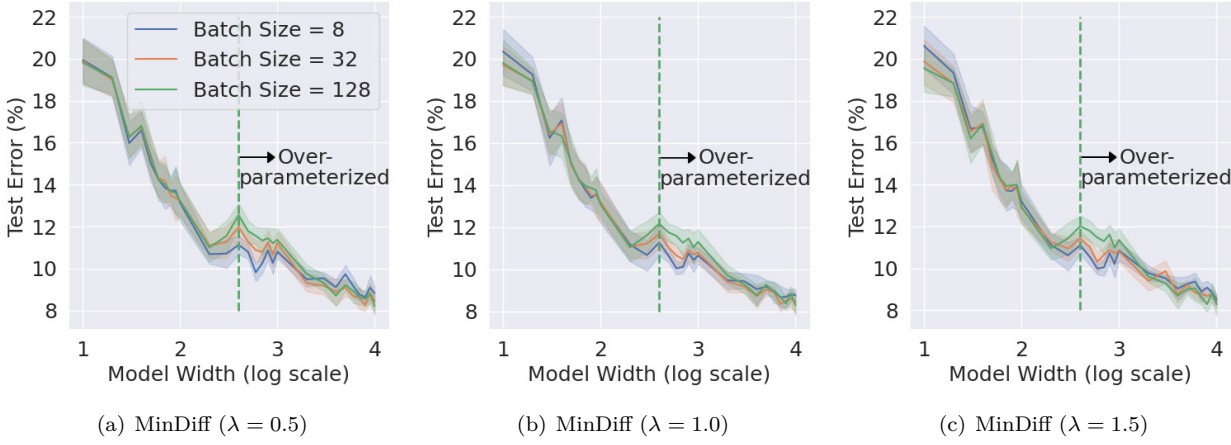

(a) MinDiff ($\lambda = 0.5$)    (b) MinDiff ($\lambda = 1.0$)    (c) MinDiff ($\lambda = 1.5$)

Figure 10: Batch size - THR with fairness constraint $< 10\%$

## C.2  All Other Regularizers

### C.2.1  Waterbirds

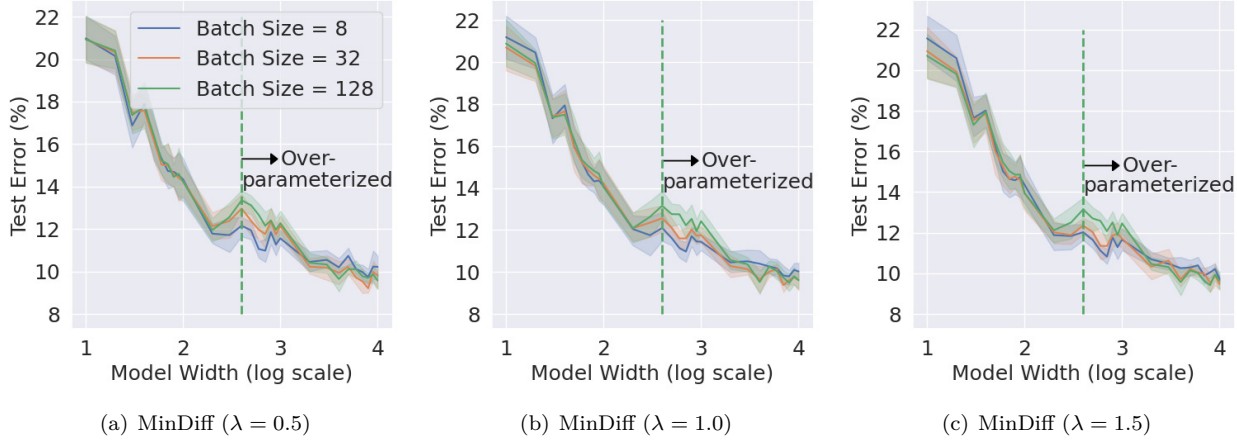

(a) MinDiff ($\lambda = 0.5$)  (b) MinDiff ($\lambda = 1.0$)  (c) MinDiff ($\lambda = 1.5$)

Figure 11: Batch size - THR with fairness constraint $< 5\%$

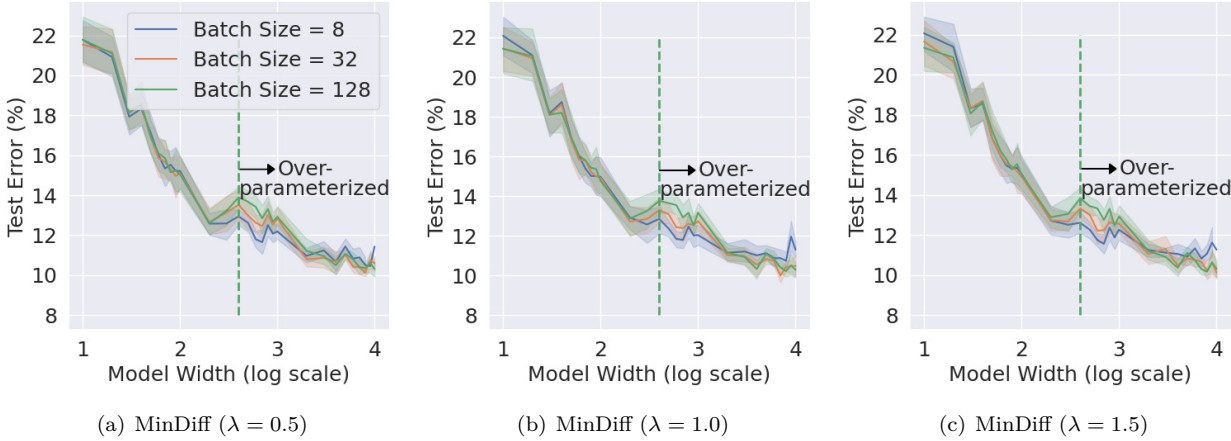

(a) MinDiff ($\lambda = 0.5$)  (b) MinDiff ($\lambda = 1.0$)  (c) MinDiff ($\lambda = 1.5$)

Figure 12: Batch size - THR with fairness constraint $< 1\%$

### C.2.2 CelebA

Table 5: We tabulate the fairness constrained test error (with a $\Delta_{FNR} \leq 10\%$ constraint) for different regularization schemes used in conjunction with MinDiff optimization ($\lambda = 0.5$) on CelebA dataset. The performance of best regularizer is highlighted in bold for each model width. Notation: wd is weight decay, es($\mathcal{L}_P$) is early stopping w.r.t primary loss and fl is flooding

| Method | Width = 5 | Width = 19 | Width = 64 |
|---|---|---|---|
| | Under- | Over- | Over- |
| $\lambda = 0$ | $5.92 \pm 0.27$ | $6.51 \pm 0.16$ | $5.76 \pm 0.14$ |
| $\lambda = 0.5$ | $6.36 \pm 0.72$ | $6.54 \pm 0.17$ | $5.88 \pm 0.19$ |
| $\lambda = 0.5 + \text{wd} = 0.001$ | $5.89 \pm 0.32$ | $6.94 \pm 0.18$ | $6.25 \pm 0.48$ |
| $\lambda = 0.5 + \text{wd} = 0.1$ | $\mathbf{5.66 \pm 0.18}$ | $\mathbf{5.27 \pm 0.16}$ | $6.40 \pm 0.18$ |
| $\lambda = 0.5 + \text{es}(\mathcal{L}_P)$ | $6.17 \pm 0.48$ | $5.91 \pm 0.37$ | $5.45 \pm 0.32$ |
| $\lambda = 0.5 + \text{fl} = 0.05$ | $6.45 \pm 0.69$ | $6.53 \pm 0.21$ | $6.16 \pm 0.09$ |
| $\lambda = 0.5+ \text{fl} = 0.1$ | $6.34 \pm 0.63$ | $5.75 \pm 0.35$ | $\mathbf{4.80 \pm 0.28}$ |

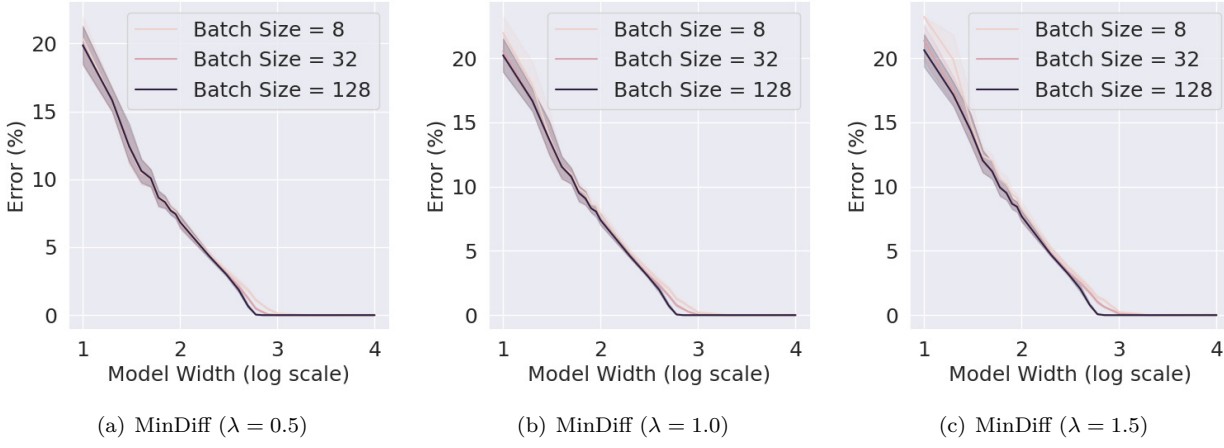

(a) MinDiff ($\lambda = 0.5$)  (b) MinDiff ($\lambda = 1.0$)  (c) MinDiff ($\lambda = 1.5$)

Figure 13: Training loss vs model widths for different batch sizes

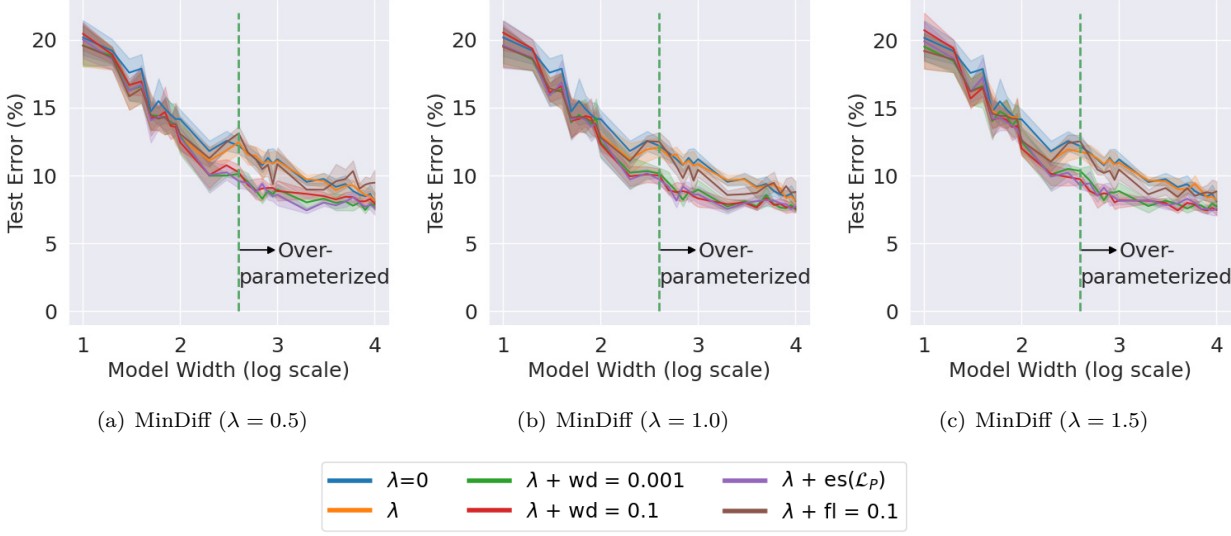

(a) MinDiff ($\lambda = 0.5$)  (b) MinDiff ($\lambda = 1.0$)  (c) MinDiff ($\lambda = 1.5$)

Figure 14: Regularization - THR with fairness constraint $< 10\%$

Table 6: We tabulate the fairness constrained test error (with a $\Delta_{FNR} \leq 10\%$ constraint) for different regularization schemes used in conjunction with MinDiff optimization ($\lambda = 1.0$) on CelebA dataset. The performance of best regularizer is highlighted in bold for each model width. Notation: wd is weight decay, $es(\mathcal{L}_P)$ is early stopping w.r.t primary loss and fl is flooding

| Method | Width = 5 | Width = 19 | Width = 64 |
|---|---|---|---|
| | Under- | Over- | Over- |
| $\lambda = 0$ | $5.92 \pm 0.27$ | $6.51 \pm 0.16$ | $5.76 \pm 0.14$ |
| $\lambda = 1.0$ | $6.25 \pm 0.61$ | $6.43 \pm 0.17$ | $6.06 \pm 0.09$ |
| $\lambda = 1.0 + \text{wd} = 0.001$ | $5.94 \pm 0.39$ | $6.75 \pm 0.35$ | $5.90 \pm 0.24$ |
| $\lambda = 1.0 + \text{wd} = 0.1$ | $\mathbf{5.80 \pm 0.17}$ | $5.55 \pm 0.19$ | $6.42 \pm 0.12$ |
| $\lambda = 1.0 + \text{es}(\mathcal{L}_P)$ | $6.32 \pm 0.40$ | $5.82 \pm 0.46$ | $4.79 \pm 0.24$ |
| $\lambda = 1.0 + \text{fl} = 0.05$ | $6.30 \pm 0.68$ | $6.43 \pm 0.17$ | $5.87 \pm 0.11$ |
| $\lambda = 1.0 + \text{fl} = 0.1$ | $6.33 \pm 0.52$ | $\mathbf{5.4 \pm 0.35}$ | $\mathbf{4.77 \pm 0.13}$ |

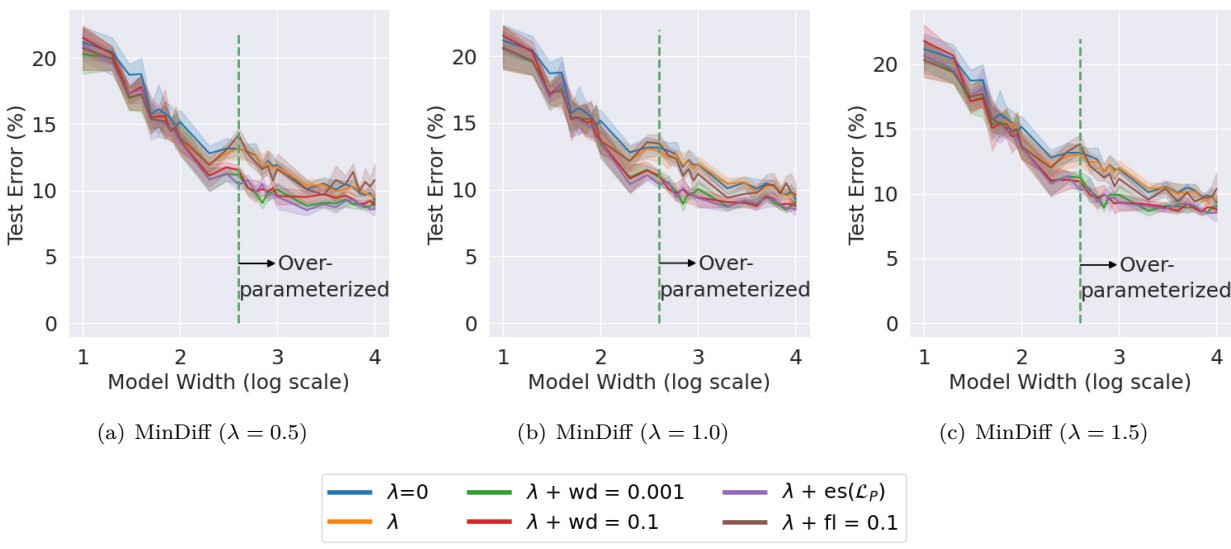

Figure 15: Regularization - THR with fairness constraint $< 5\%$

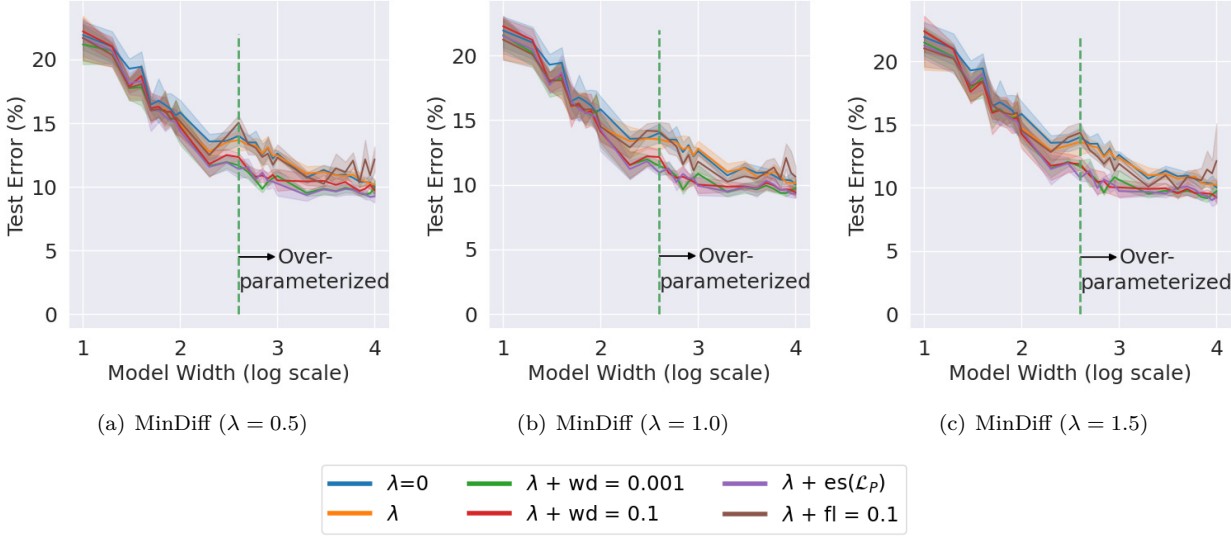

Figure 16: Regularization - THR with fairness constraint $< 1\%$

Table 7: CelebA (MinDiff: 0.5, FNR Gap $\leq 5\%$, Batch Size: 128)

| Method | Width = 5 | Width = 19 | Width = 64 |
|---|---|---|---|
| | Under- | Over- | Over- |
| Original Model | $6.22 \pm 0.31$ | $6.81 \pm 0.21$ | $6.10 \pm 0.14$ |
| MinDiff | $6.72 \pm 0.70$ | $6.78 \pm 0.16$ | $6.18 \pm 0.25$ |
| MinDiff + wd = 0.001 | $6.20 \pm 0.35$ | $7.29 \pm 0.21$ | $6.51 \pm 0.55$ |
| MinDiff + wd = 0.1 | $\mathbf{5.84 \pm 0.13}$ | $\mathbf{5.51 \pm 0.15}$ | $6.70 \pm 0.14$ |
| MinDiff + es | $6.48 \pm 0.48$ | $6.14 \pm 0.38$ | $5.61 \pm 0.30$ |
| MinDiff + fl = 0.05 | $6.73 \pm 0.70$ | $6.80 \pm 0.21$ | $6.47 \pm 0.11$ |
| MinDiff + fl = 0.1 | $6.65 \pm 0.67$ | $5.99 \pm 0.37$ | $\mathbf{5.03 \pm 0.29}$ |

Table 8: CelebA (MinDiff: 1.0, FNR Gap $\leq 5\%$, Batch Size: 128)

| Method | Width = 5 | Width = 19 | Width = 64 |
|---|---|---|---|
| | Under- | Over- | Over- |
| Original Model | $6.22 \pm 0.31$ | $6.81 \pm 0.21$ | $6.10 \pm 0.14$ |
| MinDiff | $6.57 \pm 0.56$ | $6.74 \pm 0.20$ | $6.35 \pm 0.07$ |
| MinDiff + wd = 0.001 | $6.26 \pm 0.45$ | $7.11 \pm 0.34$ | $6.21 \pm 0.23$ |
| MinDiff + wd = 0.1 | $\mathbf{6.15 \pm 0.18}$ | $5.82 \pm 0.14$ | $6.70 \pm 0.14$ |
| MinDiff + es | $6.60 \pm 0.40$ | $6.15 \pm 0.50$ | $5.05 \pm 0.18$ |
| MinDiff + fl = 0.05 | $6.57 \pm 0.70$ | $6.74 \pm 0.14$ | $6.15 \pm 0.15$ |
| MinDiff + fl = 0.1 | $6.58 \pm 0.43$ | $\mathbf{5.67 \pm 0.38}$ | $\mathbf{4.92 \pm 0.06}$ |

Table 9: CelebA (MinDiff: 1.5, FNR Gap $\leq 5\%$, Batch Size: 128)

| Method | Width = 5 | Width = 19 | Width = 64 |
|---|---|---|---|
| | Under- | Over- | Over- |
| Original Model | $6.22 \pm 0.31$ | $6.81 \pm 0.21$ | $6.10 \pm 0.14$ |
| MinDiff | $6.65 \pm 0.46$ | $6.90 \pm 0.30$ | $6.40 \pm 0.26$ |
| MinDiff + wd = 0.001 | $\mathbf{6.15 \pm 0.41}$ | $6.85 \pm 0.30$ | $5.33 \pm 0.21$ |
| MinDiff + wd = 0.1 | $6.45 \pm 0.18$ | $6.11 \pm 0.13$ | $6.81 \pm 0.10$ |
| MinDiff + es | $6.73 \pm 0.43$ | $5.89 \pm 0.45$ | $4.88 \pm 0.05$ |
| MinDiff + fl = 0.05 | $6.58 \pm 0.39$ | $6.64 \pm 0.40$ | $5.77 \pm 0.18$ |
| MinDiff + fl = 0.1 | $6.68 \pm 0.44$ | $\mathbf{5.56 \pm 0.41}$ | $\mathbf{4.86 \pm 0.14}$ |

Table 10: CelebA (MinDiff: 0.5, FNR Gap $\leq 1\%$, Batch Size: 128)

| Method | Width = 5 | Width = 19 | Width = 64 |
|---|---|---|---|
| | Under- | Over- | Over- |
| Original Model | $6.57 \pm 0.32$ | $7.13 \pm 0.22$ | $6.31 \pm 0.12$ |
| MinDiff | $6.96 \pm 0.72$ | $7.07 \pm 0.14$ | $6.50 \pm 0.26$ |
| MinDiff + wd = 0.001 | $6.52 \pm 0.35$ | $7.57 \pm 0.24$ | $6.86 \pm 0.55$ |
| MinDiff + wd = 0.1 | $\mathbf{6.12 \pm 0.13}$ | $\mathbf{5.82 \pm 0.18}$ | $6.94 \pm 0.20$ |
| MinDiff + es | $6.71 \pm .048$ | $6.41 \pm 0.42$ | $5.85 \pm 0.29$ |
| MinDiff + fl = 0.05 | $7.04 \pm 0.70$ | $7.09 \pm 0.24$ | $6.78 \pm 0.10$ |
| MinDiff + fl = 0.1 | $6.94 \pm 0.67$ | $6.20 \pm 0.42$ | $\mathbf{5.17 \pm 0.28}$ |

Table 11: CelebA (MinDiff: 1.0, FNR Gap $\leq 1\%$, Batch Size: 128)

| Method | Width = 5 | Width = 19 | Width = 64 |
|---|---|---|---|
| | Under- | Over- | Over- |
| Original Model | $6.57 \pm 0.32$ | $7.13 \pm 0.22$ | $6.31 \pm 0.12$ |
| MinDiff | $6.76 \pm 0.54$ | $7.00 \pm 0.18$ | $6.55 \pm 0.14$ |
| MinDiff + wd = 0.001 | $6.54 \pm 0.46$ | $7.43 \pm 0.41$ | $6.46 \pm 0.20$ |
| MinDiff + wd = 0.1 | $\mathbf{6.43 \pm 0.22}$ | $6.11 \pm 0.12$ | $6.97 \pm 0.17$ |
| MinDiff + es | $6.86 \pm 0.42$ | $6.40 \pm 0.53$ | $5.24 \pm 0.21$ |
| MinDiff + fl = 0.05 | $6.87 \pm 0.66$ | $7.04 \pm 0.17$ | $6.33 \pm 0.15$ |
| MinDiff + fl = 0.1 | $6.90 \pm 0.41$ | $\mathbf{5.94 \pm 0.33}$ | $\mathbf{5.09 \pm 0.08}$ |

Table 12: CelebA (MinDiff: 1.5, FNR Gap $\leq 1\%$, Batch Size: 128)

| Method | Width = 5 | Width = 19 | Width = 64 |
|---|---|---|---|
| | Under- | Over- | Over- |
| Original Model | $6.57 \pm 0.32$ | $7.13 \pm 0.22$ | $6.31 \pm 0.12$ |
| MinDiff | $6.86 \pm 0.49$ | $7.24 \pm 0.31$ | $6.64 \pm 0.26$ |
| MinDiff + wd = 0.001 | $\mathbf{6.42 \pm 0.45}$ | $7.13 \pm 0.30$ | $5.68 \pm 0.25$ |
| MinDiff + wd = 0.1 | $6.75 \pm 0.22$ | $6.29 \pm 0.17$ | $7.05 \pm 0.14$ |
| MinDiff + es | $6.97 \pm 0.38$ | $6.12 \pm 0.49$ | $5.07 \pm 0.06$ |
| MinDiff + fl = 0.05 | $6.85 \pm 0.39$ | $6.92 \pm 0.42$ | $5.93 \pm 0.24$ |
| MinDiff + fl = 0.1 | $6.89 \pm 0.45$ | $\mathbf{5.84 \pm 0.42}$ | $\mathbf{5.04 \pm 0.16}$ |

