# OpenReview forum: "Fairness via In-Processing in the Over-parameterized Regime: A Cautionary Tale with MinDiff Loss"
_TMLR — Accepted by TMLR_

### Review · Reviewer_ziUC · 2023-02-22

**Summary Of Contributions:**

The authors study the effectiveness of MinDiff for mitigating error disparity across subgroups in overparameterized models. Using experiments on CelebA and Waterbitrds in ResNet18 with varying widths, they argue that MinDiff alone is ineffective when the model is overparameterized but its effectiveness can be enhanced with regularization techniques, such as weight decay and flooding. The authors focus on equality of opportunity as a metric but the conclusions should also hold for any metrics based on error disparities, such as equalized odds, but not (for example) demographic parity.

**Audience:**

Yes

**Claims And Evidence:**

Yes

**Requested Changes:**

- Typo in Page 4: "once the validation loss has increased"  --> "once the validation loss has not increased".
- The authors should include experiments with $\lambda\in{4, 8, 16\}$. If this is a challenge timewise, please report results with $\lambda=16$.
- Can you please clarify why the setups used in the two datasets are different? It seems that the authors only have access to GPUs for CelebA but not for Waterbirds. Is this the reason?
- In Figure 3, can you also add within the same figure the loss and FNR gap for test data? This might help understand the effect of early stopping.
- What does $\lambda$ in the legend of Figure 6 mean? I think this is a typo.
- The line space in Page 10 is different from the rest of the paper.
- In Page 2, it is mentioned that Pham et al. (2021) observed that overparameterized models improve the worst case loss. This has also been recently demonstrated in ViT-22B [3]. It is also observed in [3] and [4] that overparamterization offers a more favorable tradeoff between bias and accuracy. Perhaps the paper would benefit from including this discussion in the related works since it touches upon the relationship between overparameterization and bias.

[3] Dehghani, Mostafa, et al. "Scaling Vision Transformers to 22 Billion Parameters." arXiv preprint arXiv:2302.05442 (2023).

[4] Alabdulmohsin, Ibrahim M., and Mario Lucic. "A near-optimal algorithm for debiasing trained machine learning models." NeurIPS, 2021.

**Strengths And Weaknesses:**

*Strengths*:
- The paper is well-written and easy to read. The authors present an important message about the pitfalls of using in-processing techniques for mitigating error disparities in overparameterized models.
- The authors consider several regularization techniques in their analysis and show (in the appendix) that their conclusions are insensitive to the choice of the post-hoc threshold level (which is always 10% in the main paper, but is varied in Appendix A).

*Weaknesses*:
- The main message is somehow known to practitioners already and does not only hold for in-processing methods but also in other techniques such as reweighting [1]. Since overparamterized models will drive the training error to zero, such techniques can be ineffective. As the authors cite in the paper, this has also been demonstrated in published works, such as [2] who also looked into regularization techniques including weight decay and early stopping.
- The authors only demonstrate their claims using equality of opportunity. While I think the conclusions should also hold for other error-based metrics, such as equalized odds, it would be good to demonstrate them as well.
- Throughout the experiments, the authors vary $\lambda$ only in the set $\{0, 0.5, 1, 1.5\}$. I think the authors should use an exponential spacing. It's likely that MinDiff would perform better if a very large value of $\lambda$ is used (because the loss now focuses on the proxy for error parity rather than on the cross entropy so the model wouldn't interpolate the training data).

[1] Byrd, Jonathon, and Zachary Lipton. "What is the effect of importance weighting in deep learning?" ICML, 2019.

[2] Sagawa, et al. "Distributionally robust neural networks." ICLR, 2020

---

### Review · Reviewer_vZV8 · 2023-02-22

**Summary Of Contributions:**


As far as I understand, this paper targets the problem of learning a classification model such as to meet fairness conditions. Presumably for the sake of simplicity the problems (datasets) considered in this work are binary with respect to class labels (two classes) and the special attribute values (two possible attribute values). The classification models are neural network models with many layers, hence deep learning. The paper proposes a training objective for the said kinds of problems, and empirical evaluations, with claims of improvements (or some better properties) with respect to the MinDiff method, the latter being a fairness-constrained training procedure made available by the TensorFlow Responsible AI Toolkit.

**Audience:**

Yes

**Claims And Evidence:**

Yes

**Requested Changes:**

Abstract:

Appears to be endorsing debatable claims such as "ability to generalize" (which has to be measured according to a suitable notion of generalization which the abstract has not specified) and the connections between things are not very clear. The written in the first parenthetical is strange because it is split by a period, could be avoided. Try without parenthesis: "This will apply to any disparity-based measures which care about errors or accuracy; while it won’t apply to demographic parity."

Sec. 1 Introduction

Line 1: " in a wide range" (insert "a")

A bit further down: " there is a growing" (replace "an" with "a")

Next para, first sentence: "Methods to ensure fairness can be broadly categorized according to the stage at which they are used:
pre-training, in-training, or post-training."

Next page, top line: Replace "Framework" with "Toolkit". It is not at all apparent that "(AI)" is a citation and hyperlink, try changing to "(Google AI)" to fix this. Then the corresponding reference item (when clicking the hyperlink) should read something like "Google AI. TensorFlow Responsible AI Toolkit. URL https://www.tensorflow.org/responsible_ai. (Accessed DD.MM.YYY)" -- Important to record the date because the content of web pages is subject to changes overtime.

Next paragraph: "We evaluate MinDiff on two datasets, Waterbirds and CelebA, that are commonly used in fairness literature;
and we observe several notes of caution." (comma after "CelebA" and "we observe")

Next sentence: The message of this paragraph is unclear, and there are incorrect descriptions such as the given description of "over-parametrized" while there is no description of the meaning of "under-parameterized" which is then being assumed to be known by the readers, or delegated for readers to figure out elsewhere. This needs to be improved. Trying to suggest how to reformulate:

"Because the success of deep learning can be attributed at least in part to the surprising ability of over-parameterized deep networks to generalize (Nakkiran et al., 2020), we begin by evaluating the relationship between model capacity and fairness with MinDiff. Recall that over-parameterized networks are those with more trainable parameters than training data, which therefore have sufficient capacity to
memorize the training dataset; while under-parameterized networks are those with less parameters than training data. The setting where the number of trainable parameters matches the amount of training data is usually referred to as the interpolation threshold. We observe that MinDiff does increase fairness for small, under-parameterized models, but is almost entirely ineffective on larger over-parameterized networks. Thus, in some cases, under-parameterized MinDiff models can have lower fairness-constrained error compared to their over-parameterized counterparts even though over-parameterized models are always better on baseline error (i.e., error on models trained without MinDiff optimization). We caution that when using MinDiff for fairness, ML practitioners must carefully choose model capacity, something which is generally unnecessary when fairness is not a concern and the goal is simply to minimize error."

Next paragraph: Suggesting to succinctly discuss the meaning of "disparity-based" for the sake of the unfamiliar readers. Same comment for the meaning of "demographic parity" which needs to be made explicit for the unfamiliar readers. Or point to suitable references.

Noting that the period inside a parenthetical is strange. Try to avoid. Suggesting how to reformulate:

"[..] thus turning off the MinDiff optimization. This will apply to any disparity-based measures which care about errors or accuracy, whereas It won’t apply to demographic parity which cares about [..]. Thus, we explore [..]"

Next: The parenthesis within parenthesis are an obstacle to clarity. Try to avoid. Try:

"Specifically, we consider two classes of regularization techniques: implicit such as batch sizing (Smith et al., 2021;
Barrett & Dherin, 2021) and early stopping (Morgan & Bourlard, 1990), and explicit such as weight decay (Krogh &
Hertz, 1992) and a recently proposed “loss flooding" method (Ishida et al., 2020). We find [..]"

Sec. 2 Related Work

First line: "in the literature" (insert "the")

"In-processing techniques (Prost et al. [..]) alter the training mechanism [..]"

Bottom of the paragraph: " similar qualitative conclusions might hold for other" (replace "will" with "might")?

Bottom of the page (last line): remove space before "Sagawa"

Next page, top paragraph, last sentence: "deep learning models trained using" (insert "learning")

Sec. 3 Methodology

Subsec.3.1: augment the title "Setup" to "Formal Setup" ?

First para, third line: Delete the coma after $P_{X,A,Y}$

Eq. (1): Could use "\log" in math mode LaTeX to produce a nice looking "log" function. Also, on the left-hand side, the notation $\mathcal{L}_P$ is confusing because the expression on the right hand side is the empirical loss which is a function of the data but not the data-generating distribution. The notation $\mathcal{L}_D$ would then be preferable (and replace throughout the paper).

Next line: "via stochastic gradient descent (SGD)."

Line after Eq. (2): "seek to achieve low error" (delete "test"). Note that "test error" could be interpreted as "error on the test set" while the expression following this phrase defines the error of $\hat{f}_\theta$ at population level.

After Eq. (3): Insert some comment about why/when it make sense to minimise $\mathrm{FNR}_\mathrm{gap}$ to achieve fairness.

Subsec.3.2, first para, fourth line: "Equation (1)" (eq. number enclosed in parentheses, same as in the displayed eq's)

"and $\mathcal{L}_M$ is  a differentiable proxy for the $\mathrm{FNR}_\mathrm{gap}$"

Next line: "importance of the fairness versus the empirical cross-entropy loss" (not "test error")

Eq. (4): Provide the mathematical definition of the right-hand side. Perhaps discuss its meaning and why it is used as the proxy in this setting.

Subsec.3.3, first para, third line: "as proposed by Hardt et al. (2016)" (use \citet{} instead of \citep{} in this case)

Two lines further down: "constraint is met and the error is minimized:" (remove "test" replace with "the" and delete period)

Eq (5): Remove the comma at the end of the eq.

Next paragraph mentions "test error" a few times. Perhaps this suggests that earlier in the paper you should clearly define what you mean by "test error" (mathematical definition) and discuss differences with "error on the test set"

Subsec.3.4, no comments here.

Sec. 4 Experimental Setup

Subsec.4.1, first para, first line: "Waterbirds is a synthetically created dataset" (insert "a")

Next para: Specify details about the random matrix $U$, e.g. independent Gaussian entries? What parameters for the Gaussian distribution? Assuming that all entries are Gaussian with the same parameters, but this needs to be specified as well.

Subsec.4.2, first para: remove the space between "women" and the footnote number.

Sec. 5 Experimental Results

Subsec.5.1: This section and related Figure 1 discuss the "test error" but in this case "test error" really actually means the empirical error rate evaluated on a test set (i.e. what I called "error on the test set" before) and once again the distinction needs to be made clear between the error notion at population level (which is distribution-dependent) and the empirical counterpart evaluated on a finite amount of data points (which is data-dependent).

Subsec.5.2 and 5.3: Similar comment.

Note that the figures necessarily show the test set error rates, while the target quantity of interest I think must be the distribution-dependent error at population level, what has been presented before as $\mathbb{P}[\hat{f}_\theta(X;\tau) \neq Y]$ and the attribute conditioned probabilities $\mathbb{P}[\hat{f}_\theta(X;\tau) = 0 \vert Y=1, A=0]$ and $\mathbb{P}[\hat{f}_\theta(X;\tau) = 0 \vert Y=1, A=1]$. These things need to be discussed to improve the clarity.


References:

Hardt et al. (2016) "Equality of opportunity in supervised learning" appears duplicate. Need to remove the duplicate.

The bibliographic information for Prost et al. (2019) is incomplete (how published?). Needs to be updated.

Same for Grother ert al. (2010), Madras et al. (2018), Martinez et al. (2020), Ngan and Grother (2015), Ryu et al. (2018), Savani et al. (2020), Wadsworth et al. (2018), Wang and Deng (2019), Wang et al. (2020), Zhang et al. (2018); there are others like these.

Check all the reference items to make sure complete details are provided. Make venue names consistent.

Some titles need capital-protecting (e.g. "Pareto")


**Strengths And Weaknesses:**


Strengths:

- The problem setting is interesting and of current relevance in potentially many applications.
- The experimental results are presented with sufficient detail to be reproduced (I think) and with discussions.
- It is believable that the paper's claims are supported by the experimental evaluations results.

Weaknesses:

- The paper has major (in my opinion) clarity/readability problems that may affect the ability of readers to understand it.
- The mathematical notations are unclear in some cases and the precise mathematical meaning of some things is missing.
- Most notably, there "test error" has two meanings that are conflated in this work, which can lead to confusions and may spread some misconceptions among the readership if this work is published as is.
- Many editorial changes need to be carried out before this paper is acceptable for publication (details below).

---

### Review · Reviewer_VRnw · 2023-03-02

**Summary Of Contributions:**

As the deep neural networks are increasingly being used in real-world applications, the fairness issue has also received increasing attention from the community. This work studies one widely used fairness algorithm, i.e,.MinDiff, and has some interesting findings. First, under-parameterized MinDiff models can even have lower error compared to their over-parameterized counterparts within specified fairness constraints. In addition, MinDiff optimization is also sensitive to choice of batch size in the under-parameterized regime, requiring time-consuming hyper-parameter searches. It is suggested to use regularization techniques such as L2, early stopping, and flooding in conjunction with MinDiff to train fair over-parameterized models. Over-parameterized models trained using MinDiff+regularization with standard batch sizes are fairer than their under-parameterized counterparts, suggesting that regularizers should be integrated into fair deep learning flows.


**Audience:**

Yes

**Broader Impact Concerns:**

No ethical concerns found.

**Claims And Evidence:**

Yes

**Requested Changes:**

Please refer to the Strengths And Weaknesses section.

**Strengths And Weaknesses:**

Advantages:
1. This paper addresses an important issue regarding fairness, exploring how the performance of mitigation algorithms changes in relation to the number of parameters in DNN models. Specifically, it investigates whether fairness algorithms are effective for over-parameterized models.
2. The study presents interesting experimental findings, including the discovery that under-parameterized MinDiff models can have lower error rates compared to their over-parameterized counterparts.



My concerns:
1. When considering fairness, it is crucial to keep in mind that datasets typically involve people. Waterbirds is a dataset typically used for domain generalization/worst group robustness. There are no people in this dataset. If the research topic is for domain generalization/worst group robustness, both Waterbirds and CelebA can be used as the testing datasets. However, it is not feasible the other way around. It is strongly recommended to use other datasets to replace Waterbirds.
2. Another significant concern is whether the study's findings can be generalized to other research studies and more practical real-world settings. The authors used a fixed pre-trained ResNet-18 model to extract a d-dimensional feature vector µ, then transformed it using a linear transformation and one RELU activation to a new feature space. The dimension of this space, along with the logistic regression classifier, determined whether the model was under-parameterized or over-parameterized. This setting is not commonly used in other research studies or real-world applications. A more practical approach would be to use a range of ResNet models, such as ResNet-18, ResNet-34, ResNet-50, ResNet-101, ResNet-110, ResNet-152, ResNet-164, and ResNet-1202. ResNet-18 could be treated as an under-parameterized model, while a shallower model like Alexnet could also be used. In contrast, deeper models like ResNet-110 and ResNet-152 would be considered over-parameterized models. It would be interesting to see whether the findings of this study hold true in this more practical real-world setting of under-parameterization and over-parameterization.
3. It is unclear whether the findings are only applicable to "MinDiff optimization" or whether they can be generalized to other fairness mitigation algorithms.

---

### Decision · Action_Editors · 2023-04-09

**Recommendation:** Accept with minor revision

**Comment:**

The reviewers appreciate the paper's interesting findings and agree that reinforcing this understanding within the research community is important. The paper addresses various concerns effectively, and the reviewers believe it will be a valuable addition to TMLR. However, a major concern is the limited generalizability of the paper's finding, as the study only focuses on one particular in-processing algorithm. The authors are advised to either evaluate additional algorithms or reposition the title and paper structure to emphasize that this is a case study with MinDiff. This revision would better align the paper's claims and generalizability.

**Audience:**

Yes, this paper is highly relevant to every researcher who works on ML fairness.

**Claims And Evidence:**

The authors investigate the effectiveness of fairness-aware in-processing training algorithms in over-parameterized deep neural networks, specifically focusing on the MinDiff algorithm. They reveal an "illusion of fairness" in the over-parameterized regime, where subgroup errors on the training set can be arbitrarily small. To address this issue, the authors explore regularization techniques, such as minibatching, L2, early stopping, and flooding.

Although the findings are convincing, the paper's generalizability is limited, as the authors only tested MinDiff. The title and writing imply broader applicability, but the results pertain specifically to one algorithm. It is recommended that the authors either evaluate additional algorithms or reposition the title and paper structure to focus on their case study with MinDiff.